# A male-killing *Wolbachia* endosymbiont is concealed by another endosymbiont and a nuclear suppressor

Kelly M. Richardson[1]☉, Perran A. Ross[1,2]☉, Brandon S. Cooper[3], William R. Conner[3], Thomas L. Schmidt[1], Ary A. Hoffmann[1,2]*

**1** School of BioSciences, Bio21 Institute, University of Melbourne, Parkville, Victoria, Australia, **2** Department of Chemistry and Bioscience, Aalborg University, Aalborg, Denmark, **3** Division of Biological Sciences, University of Montana, Missoula, Montana, United State of America

☉ These authors contributed equally to this work.
* ary@unimelb.edu.au

**Data Availability Statement:** All non-molecular data are available in the paper and from https://doi.org/10.26188/21862119.v1. Molecular data are available from https://melbourne.figshare.com/

## Abstract

Bacteria that live inside the cells of insect hosts (endosymbionts) can alter the reproduction of their hosts, including the killing of male offspring (male killing, MK). MK has only been described in a few insects, but this may reflect challenges in detecting MK rather than its rarity. Here, we identify MK *Wolbachia* at a low frequency (around 4%) in natural populations of *Drosophila pseudotakahashii*. MK *Wolbachia* had a stable density and maternal transmission during laboratory culture, but the MK phenotype which manifested mainly at the larval stage was lost rapidly. MK *Wolbachia* occurred alongside a second *Wolbachia* strain expressing a different reproductive manipulation, cytoplasmic incompatibility (CI). A genomic analysis highlighted *Wolbachia* regions diverged between the 2 strains involving 17 genes, and homologs of the *wmk* and *cif* genes implicated in MK and CI were identified in the *Wolbachia* assembly. Doubly infected males induced CI with uninfected females but not females singly infected with CI-causing *Wolbachia*. A rapidly spreading dominant nuclear suppressor genetic element affecting MK was identified through backcrossing and subsequent analysis with ddRAD SNPs of the *D. pseudotakahashii* genome. These findings highlight the complexity of nuclear and microbial components affecting MK endosymbiont detection and dynamics in populations and the challenges of making connections between endosymbionts and the host phenotypes affected by them.

## Introduction

Male-killing (MK) phenotypes associated with endosymbionts were first investigated in ladybugs and butterflies [1]. While MK endosymbionts often occur at low frequencies in populations, they can persist and spread through horizontal transmission or by providing a fitness advantage, such as through resource allocation or the avoidance of sib mating [2,3]. They can also invade populations with endosymbiont strains that cause cytoplasmic incompatibility (CI) as long as they are compatible with the CI strain [4]. In *Drosophila*, several male-killers

articles/dataset/D_pseudotakahashii_ddRADseq/21310478 and Genbank (accession number NZ_JAPJVH010000000). Numbers in figures can be found at https://doi.org/10.26188/21892974.v1, https://doi.org/10.26188/21863961.v1, and https://doi.org/10.26188/21862119.v1.

**Funding:** This research was supported by an Australian Research Council (https://www.arc.gov.au) Discovery grant DP120100916 to AAH, as well as a National Institutes of Health (https://nigms.nih.gov) MIRA grant R35GM124701 and a National Science Foundation (https://beta.nsf.gov) CAREER grant 2145195 to BSC. The funders had no role in study design, data collection and analysis, decision to publish, or preparation of the manuscript.

**Competing interests:** The authors have declared that no competing interests exist.

**Abbreviations:** CI, cytoplasmic incompatibility; CNV, copy number variant; Cp, crossing point; ddRADseq, double-digest restriction-site–associated DNA sequencing; MK, male killing.

associated with *Wolbachia* and *Spiroplasma* endosymbionts have been described [5,6]. However, their incidence in this genus is likely to be underestimated, partly because they can be uncommon in populations compared to CI strains that are often at a high frequency [7].

Male-killers in *Drosophila* typically result in embryo death; this includes MK associated with both *Spiroplasma* [8] as well as *Wolbachia* [6,7,9] endosymbionts. Typically, such male-killers are detected by a reduction in hatch rate coupled with changes in sex ratio; this can be one reason for their underappreciation in natural populations given that male-killers are not maintained in stocks when males are required to produce offspring [10]. However, while reproductive effects of *Wolbachia* involving CI and MK are typically mediated through effects on embryonic development that results in a loss of egg hatch, they may also be affected by sex-specific mortality later in development. For example, in mites and thrips, CI associated with *Wolbachia* has been reported as involving postembryonic mortality [11,12] and in planthoppers, mortality can occur late in development [13].

Although male-killers can result in all-female broods, there is variability in sex-ratio effects in some MK systems. In *Drosophila innubila*, *Wolbachia* density can vary among females which in turn correlates with female-biased offspring ratios, an effect that also has an epigenetic component and could contribute to stability of this infected system [5,6]. Moreover, while some male-killer phenotypes can be stable across long time periods with little resistance to them evolving over thousands of years [14], MK phenotypes associated with endosymbionts can also be suppressed by nuclear genes.

A well-documented example of MK suppression is in the butterfly *Hypolimnas bolina*, where nuclear suppression of MK revealed a CI phenotype [15,16]. In this case, a high frequency of MK in a population that persisted for many years [17] was expected to produce strong selection for a nuclear suppressor because of the fitness advantage of rare males required for offspring production [18]. Rapid recovery of male production for male-killers associated with endosymbionts has also been documented in other systems including lacewings [19] and planthoppers [20]. The genetic basis of nuclear suppression is still unclear although in *Hypolimnas bolina* it involves a single chromosomal region [21] and in suppression generated following lab-based hybridization between 2 *Drosophila* species it is polygenic [22].

In *Drosophila pseudotakahashii*, Richardson and colleagues [23] described a CI *Wolbachia* infection (*w*Pse) present at a high incidence in natural populations and causing strong CI; however, the CI was weaker in older males from which the infection could be absent despite its high density in all females. Our analyses here demonstrate that CI-causing *w*Pse is a Group-A *Wolbachia*, that is outgroup to a clade containing both *w*Mel-like and *w*Ri-like variants (S1 Fig, [24,25]). Here, we describe a second *Wolbachia* strain in *D. pseudotakahashii* that is present at a low frequency and occurs alongside the CI strain where it causes MK. Unusually, male death occurred only after embryo development and was modifiable through a nuclear gene that segregated in some laboratory lines where it increased in frequency to the extent that sex ratio reverted. We use molecular approaches to characterize the MK strain that differs for some genomic regions to the coinhabiting CI strain but is identical in other regions. We explore the presence and phylogenetic relationships of several functionally relevant regions of the *Wolbachia* genome: the *wmk* gene linked to MK [26], loci known to cause CI (*cifs*) [27,28], and the broader WO prophage regions that house these loci [29]. We also identify the genomic region associated with nuclear suppression through segregating crosses using the newly sequenced *D. takahashii* genome [30]. Our findings raise the issue of whether male-killers have been much more common in natural populations than previously assumed given that they are unlikely to be detected in the presence of common CI infections and may be affected by nuclear suppressors.

## Results

### A rare *Wolbachia* strain causes female-biased sex ratios

We established 188 *D. pseudotakahashii* isofemale lines from collections in Nowra, south-eastern Queensland, and northern Queensland, Australia. Of these, 3.72% (*N* = 7) were found to have only female F1 offspring (Table 1), and there was no significant difference in the incidence of female-biased lines across the collection sites (G = 3.87, df = 6, *P* = 0.769). No female-biased lines were found in the collections in northern Queensland or in a few individuals from Moorland.

Sequences of individuals from female-biased and non-female–biased lines using the *Pgi* and *CO1* nuclear and mitochondrial markers showed almost no variation while sequences of the *Ddc* nuclear marker showed only a small amount of variation, and for all genes there was no separation of female-biased and non-female–biased lines (sequences in Genbank (accession number NZ_JAPJVH010000000)). These results, in addition to morphological examination of occasional males emerging from the female-biased lines, supported the conclusion that the female-biased lines are indeed *D. pseudotakahashii*.

Nucleotide sequences from multiple female-biased lines were obtained for the 5 *Wolbachia* MLST loci and *wsp* [31]. Sequences presented a series of double peaks interspersed with sections without double peaks. These patterns were identical for the forward and reverse primers and present for all primer types and samples. Upon investigation, the "background" sequence was the same as the *w*Pse CI [23] strain while the double peaks presented evidence for a second strain sharing many bases in common with *w*Pse. We designed strain-specific primers and screened a subset of the isofemale lines (*N* = 111) using standard PCR. Only the female-biased lines amplified with the MK primers, suggesting that the lines with female-only offspring were indeed the only lines with the double infection. Further genomic analyses (outlined below) confirmed the presence of 2 *Wolbachia* strains.

Treating copies of the female-biased lines with tetracycline resulted in emergence of male progeny and sex ratios that were closer to 50:50 compared to copies of the lines that were not treated with tetracycline (Table 2). RT-PCR with *wsp_validation* primers confirmed their uninfected status, and these lines became self-sustaining and no longer required the introduction of males from other lines, suggesting that the female-bias is indeed related to *Wolbachia* infection.

### A double *Wolbachia* infection causes late MK

We assessed sex-ratio distortion by the *Wolbachia* double infection and its maternal transmission through crosses. We first crossed females from female-biased lines $N101^{MK}$ and $B302^{MK}$

**Table 1. Percentage of female-biased lines in the isofemale lines set up from field populations of *D. pseudotakahashii*.**

| Collection area | *N* | % Female-biased |
|---|---|---|
| **New South Wales** | **37** | **2.79** |
| Nowra | 34 | 2.94 |
| Moorland | 3 | 0.00 |
| **South-eastern Queensland** | **128** | **4.69** |
| Mount Tamborine | 74 | 4.05 |
| Cedar Creek | 10 | 0.00 |
| Mount Glorious | 44 | 6.82 |
| **Northern Queensland** (several locations, pooled) | **23** | **0.00** |

Details of collections are provided in S1 Table. *N* is the number of isofemale lines.

**Table 2. Sex ratios of female-biased lines over the first 10 generations of laboratory maintenance.**

| Line | Phenotype | F1 | F4 % Females (N) | F10 % Females (N) |
|---|---|---|---|---|
| $B246^{MK}$ | female-biased | female-only | 99.19*** (143) | 54.41 (136) |
| $B256^{MK}$ | female-biased | female-only | 93.57*** (311) | 67.92*** (159) |
| $B280^{MK}$ | female-biased | female-only | 86.01*** (193) | 60.00 (95) |
| $B289^{MK}$ | female-biased | female-only | 87.17*** (226) | 56.30 (135) |
| $B302^{MK}$ | female-biased | female-only | 91.01*** (178) | 69.03*** (113) |
| $B305^{MK}$ | female-biased | female-only | 88.57*** (175) | 66.45*** (155) |
| $N101^{MK}$ | female-biased | female-only | 96.43*** (392) | 66.36*** (*330*) |
| $B116^{CI}$ | CI | males and females | | 54.32 (81) |
| $N51^{CI}$ | CI | males and females | | 48.48 (99) |
| $B302-$ | tetracycline | N/A | | 50.00 (88) |
| $B305-$ | tetracycline | N/A | | 46.88 (128) |
| $N101-$ | tetracycline | N/A | | 55.18 (299) |

Sex ratios are compared to non-female–biased and tetracycline-cured lines where $N$ is the number of flies scored.

***Denotes a deviation from a 50:50% male: female with chi-square tests (***: $P < 0.001$).

with males from the non-female–biased $B116^{CI}$ and $N51^{CI}$ lines ($N = 15$). Three lines originating from $B302^{MK}$ females produced both males and females (% female offspring of 16.67%, 20.00%, and 42.86%); however, the remainder of crosses had all female progeny. Female offspring from female-only lines (denoted by "$MK1$") and a line derived from $B302^{MK}$ that produced male and female offspring ("$MK2$") were chosen for a second set of crosses. Note that we use italics to designate lines expressing MK and CI phenotypes associated with *Wolbachia*.

When $MK1$ females were crossed with males from the *CI* lines ($B116^{CI}$ and $N51^{CI}$), uninfected line (*TPH35-*) or the mixed sex line (*MK2*), egg hatch proportions were somewhat lower than those from crosses involving females carrying CI-causing *Wolbachia* (Table 3) although differences between the 5 crosses were marginally nonsignificant (Kruskal–Wallis test, H = 8.01, df = 4, $P = 0.072$). However, the mean percent egg-to-adult viability was half that of the control crosses and this variable differed among the treatments (Kruskal–Wallis test, H = 16.86, df = 4, $P = 0.002$). Sex ratio in the progeny also differed significantly (Kruskal–Wallis test, H = 59.09, df = 4, $P < 0.001$) and progeny from MK females were almost all female (Table 3). This suggests that MK is occurring, and mostly at a later time point than expected based on studies in other species (e.g., *D. pandora* [7]).

When females from the mixed sex $MK2$ line were crossed with *CI* males, males emerged from 5 of the 11 replicates (% female progeny of the 5 replicates ± SD = 53.63% ± 11.91), while the other 6 replicates had 100% female progeny, suggesting that the leakiness in MK in the

**Table 3. Crosses between individuals from *MK*, *CI*, and uninfected (-) lines.**

| Cross | N | Mean eggs laid ± SD | Mean % egg hatch based on egg score ± SD | Mean % eggs to adult ± SD | % Female progeny ± SD |
|---|---|---|---|---|---|
| ♀*MK1* x ♂*CI* | 53 | 21.34 ± 10.02 | 79.53 ± 18.88 | 44.32 ± 22.40 | 100 |
| ♀*MK2* x ♂*CI* | 11 | 20.73 ± 10.35 | 77.12 ± 29.12 | 51.33 ± 23.05 | 76.81 ± 25.70 |
| ♀*MK1* x ♂*-* | 4 | 22 ± 8.37 | 87.54 ± 10.02 | 51.57 ± 20.15 | 98.21 ± 3.57 |
| ♀*MK1* x ♂*MK2* | 6 | 18.83 ± 6.97 | 79.38 ± 20.26 | 37.23 ± 11.76 | 100 |
| ♀*CI* x ♂*CI* | 10 | 17.80 ± 8.83 | 94.47 ± 9.82 | 81.43 ± 28.58 | 58.42 ± 12.77 |

*MK1* females are from parents producing female-only offspring and *MK2* females are from parents producing both male and female offspring. $N$ is the number of replicates after those that did not mate and those that had <7 eggs were removed.

female parent was not passed on to all her daughters. Mean % egg to adult viability (± *SD*) for the 5 crosses with males emerging was 68.31% ± 7.83 compared to 46.26 ± 13.00 for the replicates with only female progeny, indicative of MK rather than feminization.

Screening indicated that all but 2 females from the *N101^MK^* and *B302^MK^* lines used in the crosses had both the MK and CI *Wolbachia* strains (*N* = 73), suggesting incomplete maternal transmission, which is common in *Drosophila* (e.g., [32,33]). Males from the *B116^CI^* and *N51^CI^* lines used in the crosses were 80% and 60% infected with the CI strain, respectively (*N* = 45 and 25, respectively), consistent with findings elsewhere [23]. All *MK2* males had the CI strain (*N* = 9), while all but one had the MK strain.

## Rapid loss of MK, but long-term stability of MK *Wolbachia* during laboratory culture

Over time, males emerged in all the *MK* lines that were not tetracycline-treated (Table 2). By F10, the lines had only a slightly female-biased sex ratio. At F11, we tested 2 males and 2 females from a range of *MK* and non-female–biased lines and found that all but one female and all males from the female-biased lines carried the MK strain (S2 Table). We tested a further 34 males from the *B289^MK^*, *B280^MK^*, and *B302^MK^* lines (*N* = 7, 8, and 19, respectively) and all but 2 carried the MK strain, suggesting that despite the appearance of males, the MK infection was still present in the female-biased lines.

After 68 generations, we again screened a subset of the lines for infection status. Despite the *MK* lines no longer having strong sex-ratio biases, all individuals tested from 5 *MK* lines were infected with the MK strain (females: *N* = 47, males: *N* = 47). Densities of the MK *Wolbachia* strain were similar between the sexes (S2 Fig, GLM: $F_{1,84}$ = 0.281, *P* = 0.597). Additionally, all females across the *CI* and *MK* lines were infected with the CI strain (*N* = 76). Consistent with previous research [23], presence of the CI strain was variable in males, with 90% of males from the *CI* lines (*N* = 30) and 89% of males from the *MK* lines (*N* = 47) carrying the CI strain, and *Wolbachia* density also being much lower in males than females (S2 Fig, $F_{1,129}$ = 145.531, *P* < 0.001). Sanger sequencing of the MLST genes from males singly infected with MK confirmed the presence of only the MK strain with alternate bases to the CI strain in locations where double peaks were present in double-infected individuals. Sequences published in Genbank (accession number NZ_JAPJVH010000000).

Several factors may contribute to the reappearance of males in these lines, including the potential effects of laboratory rearing on the expression of sex ratio distortion. In 2 lines that we examined in detail (see below), we show that one of the main reasons was likely to have been the emergence of nuclear-based suppression of the MK phenotype.

## Genomic analyses point to a single CI and double MK *Wolbachia* infection

Table 4 presents assembly statistics for draft *Wolbachia* assemblies for the *Smith+* line (an infected CI line—see S3 Table) and the *N101^MK^* line (an infected MK line—see S3 Table) with

**Table 4. Assembly statistics for *Smith+* and *N101^MK^*, with the complete *w*Mel genome for reference.**

| Assembly stats | Smith+ | N101^MK^ | wMel |
| --- | --- | --- | --- |
| Total length | $1.31*10^6$ | $1.57*10^6$ | $1.45*10^6$ |
| Number of scaffolds | 1 | 145 | 1 |
| BUSCO (Single|Duplicated|Fragment|Missing) | 180|0|3|38 | 161|17|3|40 | 180|0|2|39 |
| Longest scaffold | Complete | 107,559 | Complete |
| N50 | Complete | 45,215 | Complete |

the complete *w*Mel genome for comparison. *Smith+* assembled into a complete, circular genome. In contrast, we could not confidently separate the 2 *Wolbachia* infecting *N101*[MK] or assemble a complete genome for either strain. The total number of genes meeting our criteria were similar for *Smith+* ($N = 180$), *N101*[MK] ($N = 178$), and for the *w*Mel reference genome ($N = 180$). BUSCO found 17 duplicated genes in *N101*[MK], compared to zero in *Smith+* and in the *w*Mel genome.

*Smith+* is a Group-A *Wolbachia* and outgroup to a clade containing *w*Ha [34], *w*Ri-like (*w*Ri and *w*Ana [35]), and *w*Mel-like (*w*Au, *w*Mel, and *w*Yak [25]) *Wolbachia* (S1 Fig). These Group-A *Wolbachia* diverged from Group-B strains like *w*Pip_Pel in *Culex pipiens* [36] and *w*No in *D. simulans* [34] up to 46MYA [37]. We could not estimate the placement of the *Wolbachia* infecting the *N101*[MK] genotype since we could not confidently separate the 2 strains.

Fig 1A and 1B show the normalized read depth of the *Smith+* and *N101*[MK] lines, respectively, across the complete *Smith+* genome. As expected, *Smith+* has almost no deviations from 1 across the bulk of the genome, with the exception of 4 windows (a total of 3,000 bp) that have a normalized read depth over 3 (Fig 1B). This could plausibly represent sequencing bias or an assembly error in a small highly repeated area, since we do not expect CNVs when comparing *Smith+* to itself. In contrast, 2 regions of the *N101*[MK] genome totaling approximately 150 kb display approximately 0.7 normalized depth from positions around 600k to 700k and 900k to 950k. For 2->1 copy number changes, we would expect 2 paired regions, which we do not observe. While possible, for a potential 3->2 change (which would have depth close to 0.7), we would expect 3 affected regions. It seems more plausible that these genomic regions are present in the CI-causing *N101*[MK] strain but not in the second *N101*[MK] strain. There may be regions only present in the second strain, but our analysis would not identify those regions.

Our search for homologs of serine recombinases revealed 3 sr3WO prophage copies in the *Smith+* assembly. All 3 of these WO copies are in genomic regions that we predicted are not present in the second *N101*[MK] strain: 2 copies in the 600k to 700k region and 1 in the 900k to 950k region (Fig 1B). The 2 copies in the 600k to 700k region are identical. We also observed 3 sr3WO copies in the *N101*[MK] assembly: 1 identical to the 2 copies in the 600k to 700k region of the *Smith+* assembly, 1 identical to the copy in the 900k to 950k region of the *Smith+* assembly, and 1 not identical to any in *Smith+*. We predicted that the distribution of WO prophages in the CI-causing *N101*[MK] *Wolbachia* is the same as *Smith+*, despite only 1 observed copy in the 600k to 700k region of our assembly. Because the *N101*[MK] assembly is fragmented, it is likely the assembly process collapsed 2 copies into a single contig. We also predicted that the additional sr3WO copy that is present in the *N101*[MK] assembly—and absent in the Smith+ assembly—is located in the MK-causing *N101*[MK] strain.

We observed CI-causing factors (*cifs*) in both the *Smith+* and *N101*[MK] assemblies [27,28]. *cifs* (*cifA/B*) are classified into 5 phylogenetic clades (Types I to V) [38,39]. The *Smith+* genome contains 1 Type 1 and 1 Type 2 *cif* pair. The Type 2 pair falls in the 600k to 700k region likely not present in the MK-causing *N101*[MK] strain, while the Type 1 pair falls outside of this region. In the *N101*[MK] assembly, there are 2 Type 1 pairs and 1 Type 2 pair. The Type 2 pair and 1 of the Type 1 pairs are identical to the versions in the *Smith+* assembly, while the second Type 1 pair is unique. Thus, we predicted that the Type 1 pairs and the Type 2 pair are in the CI-causing *N101*[MK] strain, and that the unique Type 1 pair is in the MK-causing *N101*[MK] strain. The $cifA_{[T1]}$ and $cifA_{[T2]}$ phylograms are presented in S3 and S4 Figs, respectively.

Our analyses also discovered homologs of *w*Mel *wmk* (a gene predicted to underlie MK, see [26]) in both the *Smith+* and *N101*[MK] draft assemblies. There are 2 *wmk* copies in our *Smith+* assembly with 100% identity to each other and with 88.1% identity to *w*Mel *wmk* across bases 27–873. However, bases 1–26 and 874–912 across the region in *Smith+* *wmk* have only 37%

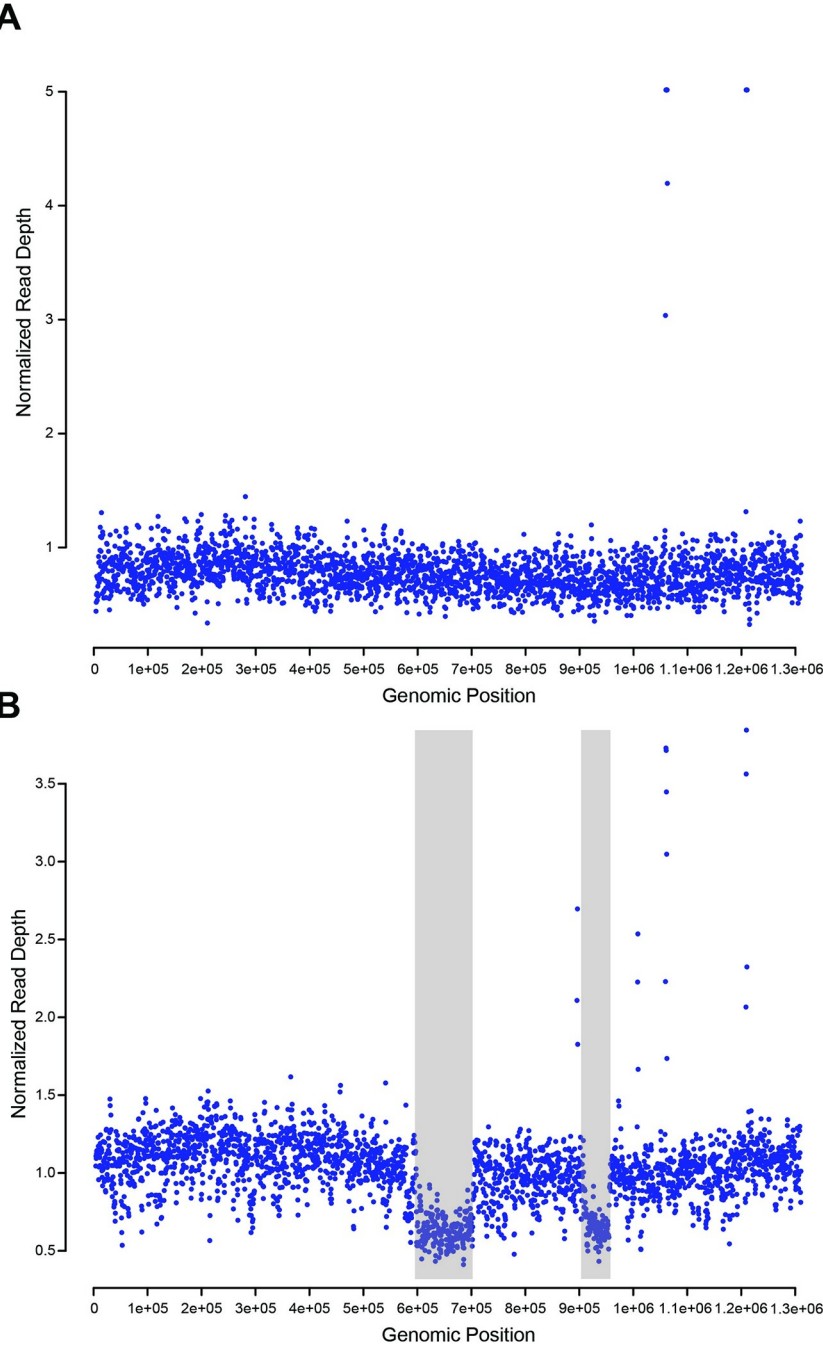

**Fig 1.** (A) Normalized read depth of Illumina *Smith+* reads across the *Smith+* genome in 1,000 bp sliding windows. (B) Normalized read depth of Illumina *N101^MK* reads across the *Smith+* genome in 1,000 bp sliding windows. Gray boxes denote regions displaying copy number variation. Depth was capped at 5 for readability. The data underlying this figure can be found in https://doi.org/10.26188/21892974.v1.

and 41% identity to *w*Mel *wmk*, respectively. We also observed a frame shift in bases 1–26, converting the third codon to a stop codon. The next potential start codon is positioned at site 186. Both *wmk* copies in our *Smith+* assembly are in the 600k to 700k region we believe is not present in the MK-causing *N101^MK* strain. In the *N101^MK* assembly, we observed 1 *wmk* copy

that is identical to the 2 identical copies in the *Smith+* assembly, plus another copy with 99.67% identity to *w*Mel *wmk* across all 912 bases. We expected the copy in the $N101^{MK}$ assembly that is identical to the copies in the *Smith+* assembly to represent 2 copies in the CI-causing $N101^{MK}$ strain due to fragmented assembly considerations described above. We predicted that the second intact copy with high identity to *w*Mel is carried by the MK-causing $N101^{MK}$ strain. All *wmk* relationships are presented in S5 Fig.

## Dominant MK suppression not linked to *Wolbachia* density

We introgressed 2 lines that maintained the MK *Wolbachia* strain but had no female bias ($N101^{MKS}$ and $B302^{MKS}$, with "*S*" in *MKS* denoting suppression of the MK phenotype) into the genetic background of a CI-only line treated with tetracycline (*B116-*) to test if the MK phenotype could be restored. Three out of 19 lines from the Nowra background reverted to MK (producing only females) after a single cross to *B116-* (Table 5), with the proportion of lines inducing MK increasing with subsequent backcrossing. Lines that produced only females continued to show a strong female bias in the following generations, with 93.75% ($N = 32$) of lines in backcross 3 and 92.31% ($n = 39$) in backcross 4 being female-only, with those producing males being highly female-biased (mean 90% female, $N = 5$). Crosses between females from the *MK* lines and males from the *MKS* lines produced only females, but their offspring included both sexes (Table 5). These results suggest dominant nuclear suppression of the MK phenotype.

To test whether MK suppression was associated with changes in *Wolbachia* density, we measured the density of the CI and MK strains in the *MK* (female-only offspring) and *MKS* (mixed-sex offspring) lines resulting from backcrossing as well as the original $N101^{MKS}$ and $B302^{MKS}$ lines (S6 Fig). We found no difference in MK *Wolbachia* density between females from the *MK* and *MKS* lines (GLM: Nowra, $F_{1,28} = 0.132$, $P = 0.719$, Brisbane, $F_{1,28} = 2.000$, $P = 0.168$), indicating that suppression of the MK phenotype is not due to a decrease in *Wolbachia* density.

We performed crosses between *MK*, *MKS*, *CI*, and uninfected lines to reassess the MK phenotype and test the ability of *MKS* males to induce cytoplasmic incompatibility. The offspring of *MK* females were strongly female-biased (Table 6), with only 2/22 replicates producing males. Despite similar egg-hatch proportions from crosses involving *MK* and *MKS* females, egg-to-adult viability of the offspring from *MK* females was half that of the other crosses. *MKS* males induced strong CI with uninfected females, with 6.17% of eggs hatching compared to

**Table 5. Segregation of MK suppression.**

| Female line | Males in backcross | Percent of lines with female-only offspring (*N* female parents tested) | | |
|---|---|---|---|---|
| | | F1 | Backcross 1 | Backcross 2 |
| $N101^{MKS}$ | $B116^-$ | 0 (20) | 15.79 (19) | 48.28 (29) |
| $B302^{MKS}$ | $B116^-$ | 0 (19) | 0 (20) | 39.29 (28) |
| $N101^{MKS}$ | $N101^{MKS}$ | 0 (20) | 0 (19) | 0 (16) |
| $B302^{MKS}$ | $B302^{MKS}$ | 0 (18) | 0 (20) | 0 (16) |
| $B116^-$ | $B116^-$ | 0 (20) | 0 (20) | 0 (17) |
| $N101^{MK}$ | $N101^{MKS}$ | 100 (9) | 4.76 (21) | 0 (17) |
| $B302^{MK}$ | $B302^{MKS}$ | 100 (9) | 0 (20) | 0 (17) |

Females were from the *MK Wolbachia* strains that either expressed (*MK*) or did not express (*MKS*) the MK phenotype or from a line lacking MK (*B116*) crossed to males from different strains to produce F1 and backcross generations.

**Table 6. Incompatibility and sex ratio in crosses between individuals from *MK*, *MKS*, *CI*, and uninfected (-) lines.**

| Cross | N | Mean % egg hatch based on egg score ± SD | Mean % eggs to adult ± SD | % Female progeny ± SD |
|---|---|---|---|---|
| ♀MK x ♂MKS | 22 | 85.86 ± 7.33 | 32.22 ± 19.89 | 96.67 ± 11.55 |
| ♀MKS x ♂MKS | 27 | 73.54 ± 28.04 | 66.04 ± 33.34 | 44.94 ± 20.36 |
| ♀MKS x ♂CI | 27 | 73.70 ± 21.47 | 66.48 ± 33.19 | 53.92 ± 24.11 |
| ♀CI x ♂MKS | 25 | 82.71 ± 15.68 | 62.50 ± 26.35 | 44.44 ± 21.69 |
| ♀- x ♂MKS | 24 | 6.17 ± 8.14 | 62.84 ± 35.50 | 50.55 ± 35.44 |
| ♀CI x ♂CI | 20 | 68.00 ± 23.49 | 70.29 ± 35.75 | 46.01 ± 17.97 |
| ♀- x ♂- | 20 | 76.00 ± 24.10 | 60.72 ± 29.84 | 54.06 ± 19.52 |

N is the number of replicates after those that did not mate and those that had <7 eggs were removed.

≥68% in the controls. Egg-hatch proportions in the *MKS* male x *CI* female cross were similar to the controls (Table 6), suggesting that the MK *Wolbachia* strain does not induce CI or has the same compatibility type as the CI strain.

## Molecular analysis of segregating lines highlights a region with a selective sweep

We performed ddRADseq on lines derived from $N101^{MK}$ and $B302^{MK}$ that produced mixed-sex offspring (*MKS*) or female-only offspring (*MK*) to identify genomic regions associated with MK suppression (Fig 2A). We identified 3 contigs of >1 Mbp length where SNPs were structured in line with MK-suppression phenotype (Fig 2B). Of these, a specific region on contig NW_025323476.1 (positions: 3,321,074 to 4,677,392) showed strongly reduced variation in the *MK* lines but normal patterns of variation in the *MKS* lines, which matched expectations of a selective sweep on the *MK* lines. This region was estimated to contain 131 unique genes and 153 gene products (see S4 Table for a list). Genome-wide heterozygosity was lower in *MK* lines than in *MKS* lines (S3 Table), though analysis of each contig in isolation showed that this difference was wholly due to heterozygosity differences in the 3 contigs from Fig 2B, where $H_O$ was 23% smaller than other contigs in *MK* lines but 49% larger than average in *MKS* lines. Backcrossing was expected to produce negative genome-wide $F_{IS}$, and this was more negative in *MKS* lines than *MK* lines. In the 3 contigs from Fig 2B, $F_{IS}$ was 84% less negative than other contigs in *MK* lines but 60% more negative in *MKS* lines.

## MK suppression can spread rapidly in mixed populations

We set up mixed populations of *MK* (female-only offspring) and *MKS* (male and female offspring) females to track the spread of MK suppression across generations. Populations with only *MKS* females showed a relatively stable 1:1 sex ratio across generations, while the populations with only *MK* females remained stably and strongly female-biased (Fig 3). The mixed populations all tended towards a 50:50 sex ratio after a few generations, reflecting the strong spreading ability of the suppressor(s). Even the population that started with 90% *MK* reverted to a 50:50 sex ratio after only a few generations. To test for any fitness benefit that might be provided to female offspring by MK *Wolbachia*, we correlated sex ratio in experimental vials to the number of females produced in the following generation and found negative associations (G1, $r = -0.304$, $P = 0.057$, $N = 40$; G2, $r = -0.804$, $P < 0.001$, $N = 40$). These results indicate that MK does not provide a detectable fitness benefit in terms of female production.

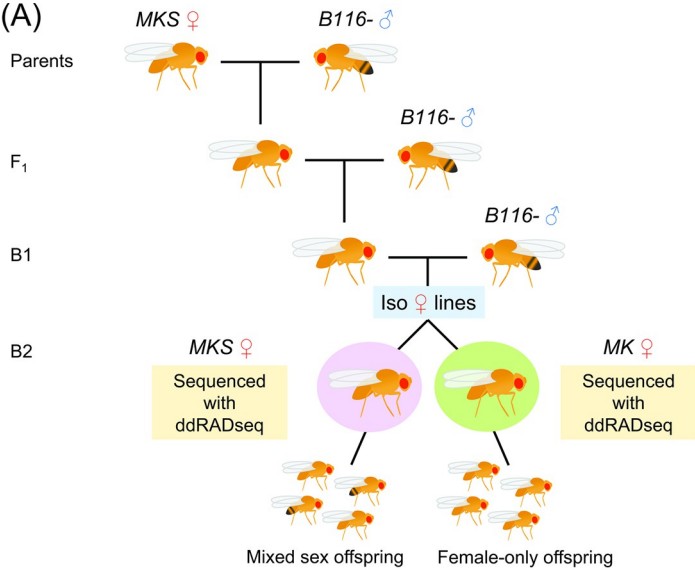

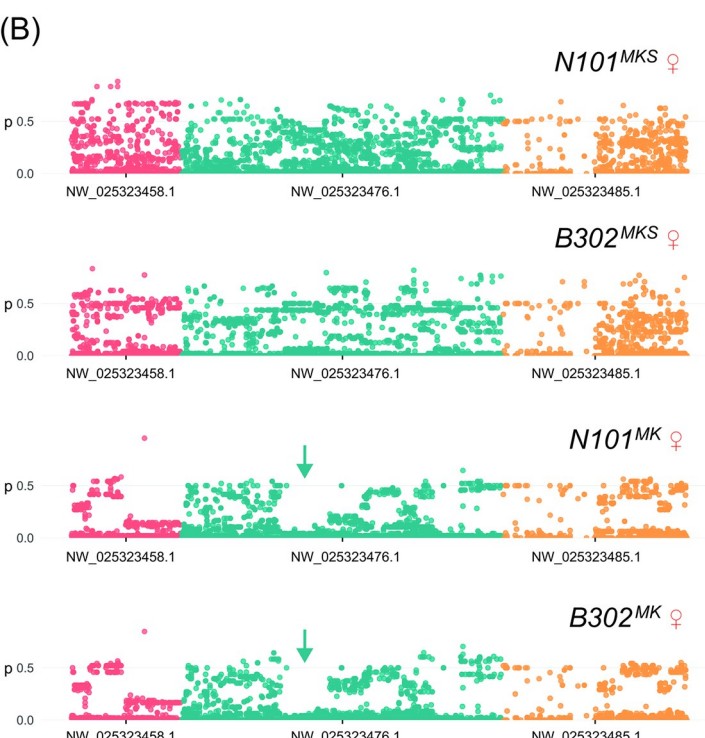

**Fig 2.** Molecular analysis of segregating lines; (A) is a cartoon detailing the experimental crosses and (B) shows frequency plots of non-reference alleles on the 3 contigs where structure followed MK suppression phenotype. A selective sweep pattern denoted by arrows is apparent on the NW_02532476.1 contig. The data underlying this figure can be found in https://doi.org/10.26188/21863961.v1.

## Discussion

### A double *Wolbachia* infection associated with MK

We report the presence of a double *Wolbachia* infection in *D. pseudotakahashii*, with the second strain inducing MK over and above the CI phenotype associated with the first strain. In

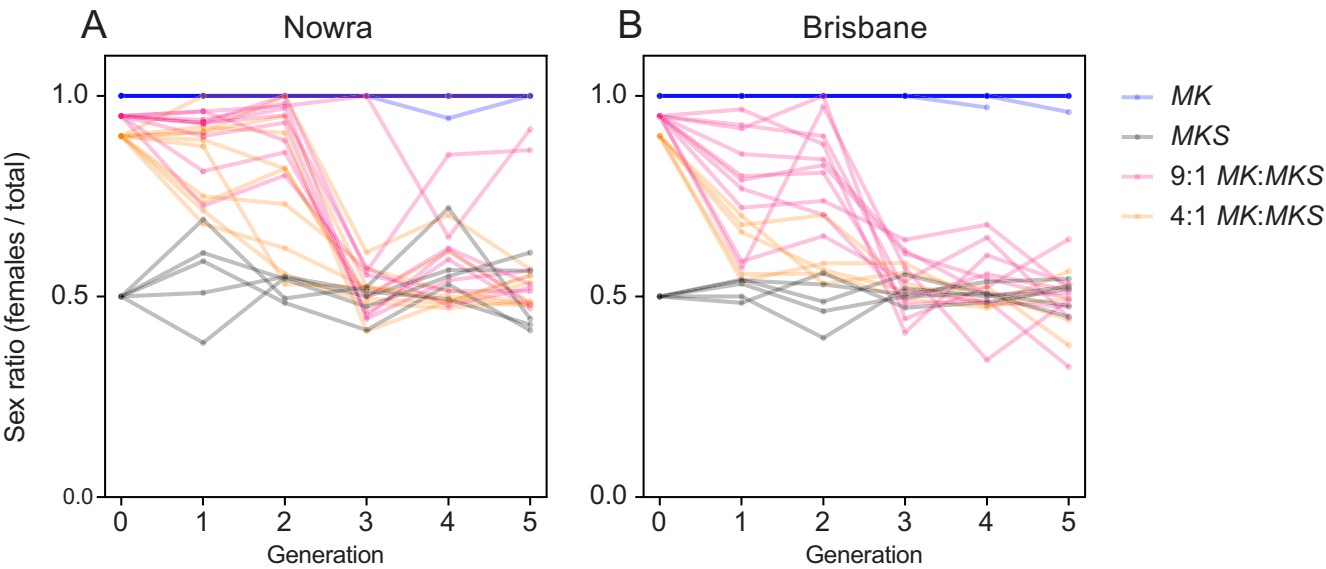

**Fig 3.** Changes in sex ratios across generations at different initial frequencies of *MK* and *MKS* females for the Nowra (A) and Brisbane (B) populations. Each line shows the sex ratio of a single replicate vial across generations Generation 0 denotes expected sex ratios based on starting ratios of *MK:MKS* females. The data underlying this figure can be found in https://doi.org/10.26188/21862119.v1.

previous studies on *Drosophila Wolbachia* infections, strains have been isolated that cause either CI or more infrequently MK, but in these cases the phenotypes are associated with strains that occur in separate hosts (e.g., [7]). Here, we have evidence for 2 strains in the same individual. This was initially based on patterns of double peaks we observed in the original MLST/*wsp* analysis supported by the subsequent genomic analysis and the *gatB* MLST sequences that were used to design primers that could distinguish the strains.

In the genomic analysis, the draft $N101^{MK}$ assembly we obtained was larger than the draft *Smith+* genome by approximately 260k bases, with BUSCO finding 17 genes duplicated in $N101^{MK}$ compared to zero in *Smith+*. While a large novel insertion or duplication in $N101^{MK}$ could potentially explain this, it seems more plausible that this pattern results from a double *Wolbachia* infection in the $N101^{MK}$ genotype. If we assume that 2 closely related *Wolbachia* infect $N101^{MK}$, as supported by preliminary analysis of MLST loci and by the normalized read depth plots across our draft genomes, similar regions sequenced from 2 unique genomic sources (i.e., 2 different *Wolbachia*) could have assembled into 1 scaffold. This would explain why the $N101^{MK}$ assembly is only 260k bases longer at 1.57 million bp while a complete *Wolbachia* genome is typically on the order of 1 to 1.4 million bp.

This does not refute the insertion or duplication hypothesis, but a copy number decrease like 2->1 or 3->2 would require multiple identical regions to be affected on the normalized depth plots. For example, 2->1 would require 2 paired regions each at 0.5 depth and 3->2 would require 3 regions at 0.66 depth, which we do not observe. Rather, the 0.7 normalized average depth could be explained by titer differences between the 2 strains. If the CI-causing *Wolbachia* in $N101^{MK}$ makes up 70% of the total *Wolbachia*, we would expect the regions only present in it to have depth of 0.7 times the rest of the genome. Future work aimed at separating these *Wolbachia* in doubly infected genotypes, or at identifying genotypes singly infected with the currently uncharacterized strain, will help confirm this.

## An uncommon infection that is nevertheless widespread

In *D. pseudotakahashii*, as in *D. pandora* and other species, the MK phenotype can be relatively uncommon and only detected when many lines from the field are screened [7]. Nevertheless, we found the MK phenotype at multiple locations across the range of *D. pseudotakahashii*, a species found in cooler habitats along eastern Australia such as high elevation sites and rainforests [40]. These findings highlight the challenges involved in accurately characterizing *Wolbachia* phenotypes from molecular surveys where phenotypes are typically determined from only a few individuals and locations and where superinfections may not be recognized, particularly when *Wolbachia* titre can depend on temperature at both the high and low extremes [41,42]. We suspect that superinfections and in particular combinations of CI and MK strains are more common than currently realized. There are currently relatively few cases where *Drosophila* species are superinfected although exceptions have been known for some time (e.g., [43]). Detection of an MK phenotype may also be less likely when MK phenotypes are only expressed at the larval stage. Crosses undertaken to characterize *Wolbachia* phenotypes may often be terminated at the egg stage when CI is normally expressed, including for CI expressed by *w*Ri group *Wolbachia* [35] which our genomic analysis shows to be closely related to the *w*Pse *Smith+* strain.

The high level of the CI strain in *D. pseudotakahashii* is presumably maintained by strong CI and high maternal transmission. Although the CI *Wolbachia* density in males is low as evident in our study and also noted previously [23], CI was strong, consistent with the presence of CI loci in the *Smith+* genome. Our analyses revealed an additional copy of the Type 1 loci in *N101^{MK}* that we predict originates from a second *Wolbachia* in this putatively doubly infected genotype. Since both Type 1 loci reside on the same contig, it seems plausible this contig contains a misassembled chimera with elements from both strains.

Based on the lack of complete transmission of the double infection as detected in our experiments (unlike the CI strain), it is hard to see how the MK phenotype would be maintained in a population unless there is a fitness advantage to MK under some situations. This advantage may not need to be large, given that MK females are fully compatible with the CI-inducing males, which can allow infections distorting sex ratio to invade populations [4]. The MK infection remains stable in many laboratory lines after long-term culture, even when the MK phenotype is lost. Moreover, there is no interaction between the density of the CI and MK strains in that the CI density is similar regardless of whether a line is also carrying an MK strain, suggesting that these strains are independent, which is further supported by the presence of a few males which have lost a detectable CI strain but where the MK strain was still found.

At this stage, the genetic basis of MK in this system and most others is unclear. We identified multiple homologs of the *wmk* gene that is predicted to contribute to MK [26]. The *wmk* homologs in the CI-causing *Smith+ Wolbachia* appear pseudogenized and are in a region we predicted to be absent in the MK-causing *N101^{MK}* strain (Fig 1B). We also observed at least 1 *wmk* copy in the *N101^{MK}* assembly identical to the 2 copies in the *Smith+* assembly. Notably, another unique copy with more than 99% identity to *wmk_{wMel}* is also present in the *N101^{MK}* assembly that we predicted to be in the MK-causing *N101^{MK}* strain.

These observations hint that *wmk* could underlie the MK we observed, although more work is needed to confirm *wmk*'s role in MK here and more broadly. While transgenic expression of *wmk* from *w*Mel in *D. melanogaster* causes MK and DNA defects that are similar to those observed during development in natural MK systems [26], *w*Mel does not naturally cause MK in *D. melanogaster*—even in the original crosses that eventually led to the identification of this infection there was no evidence of a skewed sex ratio [44]. Host suppression of MK such as observed in other systems could potentially explain this pattern in the *w*Mel-*D. melanogaster*

system. However, *w*Mel seems to not cause MK in any host backgrounds, including in divergent *Aedes aegypti* where *w*Mel has been successfully applied to controlling the spread of dengue and other viruses [45]. The *wmk* homologs derived from other *Wolbachia* also often kill males *and* females when overexpressed transgenically in *D. melanogaster* [46], raising questions about the specificity of *wmk* to male development. Our discovery here of a *wmk* copy closely related to $wmk_{wMel}$ in an MK *Wolbachia* strain will assist future work aimed at unraveling the contribution of *wmk* to MK.

Other loci and mechanisms could be involved in MK, including Oscar in the expression of MK in 2 lepidopteran insects via inhibition of a protein that is essential for both masculinization and dosage compensation [47]. Our assemblies do not include any homologs of Oscar, which may not be surprising given that Oscar functions in a *ZW* system. Other mechanisms involved might include pathogenic effects of over-replicating *Wolbachia* that might have male-specific effects if the over-replication is centered on male tissue. Pathogenic effects of *Wolbachia* have been described previously, including the *w*MelPop infection of *D. melanogaster* that shortens lifespan [48] and the introduced *w*AlbB strain in *Aedes aegypti* that can cause female infertility [49]. Mechanisms involved in MK could be studied further by developing a way of sexing early instar larvae of *D. pseudotakahashii* and then investigating changes in male larvae prior to their death.

## A rare case of MK suppression

In our study, we found a rapid increase in the incidence of males in lines that were initially heavily skewed towards females. This observation may reflect several factors including a potential impact of laboratory rearing on the expression of the MK phenotype, but our experimental work showed that MK suppression was an important factor in the lines we examined in detail. Few cases of MK suppression have been documented [50], and these have largely been uncovered through introgression experiments and transinfections. Here, we observed the rapid spread of MK suppression when female-biased lines were brought into the laboratory, but find no evidence of MK suppression in natural populations. It is possible that both the expression of MK and suppression are environmentally dependent, or that the advantages provided by MK are not selected for under laboratory conditions, such as when there is a strong level of sib competition which could favor the MK phenotype [51]. This situation contrasts with MK in another *Drosophila* (*D. innubila*) where no resistance to MK appears to have evolved over thousands of years [14].

In our study, MK was restored through introgression into a background with no suppression alleles, but suppression spread quickly when the suppression allele was present at low frequencies in mixed populations. This is not surprising because nuclear suppression alleles would be spread by both males and females—given that female *D. pseudotakahashii* with MK *Wolbachia* still need to mate and offspring would then acquire suppression alleles that show dominance based on our backcrossing results. Although we have yet to identify the gene(s) involved, we have made progress in locating it to a chromosomal region where there are several candidates which now require finer scale mapping.

At this stage, it is unclear why MK suppression is rare (or perhaps unexpressed) in natural populations of *D. pseudotakahashii*. Suppressor alleles would be expected to have a fitness advantage if MK is at a high frequency and this results in a low incidence of males in a population [18]. MK suppressors have been shown to spread rapidly in field populations of both butterflies [18] and lacewings [19]. However, there may be a cost associated with these genes in the absence of MK, resulting in populations remaining polymorphic for these genes particularly when MK is favored under some conditions. Polymorphism for MK suppression has

been noted in other systems including ladybirds [52]. Unfortunately, we lack information on the mechanistic basis of MK suppression to understand potential costs, but the *D. pseudotakahashii* system may provide a model system for investigating this further.

## Concluding remarks

Although the diverse phenotypic effects of *Wolbachia* and other endosymbionts have been recognized for some time [53], we have only recently started to make progress in understanding the population dynamics of multiple endosymbiont strains within the same individual and the interplay between the phenotypic effects associated with endosymbionts and the nuclear genome. We show here that this is a rich area for further analyses, particularly when coupled with recent advances in *Wolbachia* genomics and understanding the *Wolbachia* genes generating the phenotypic effects. Building this understanding is particularly important as endosymbionts start to be used in applied contexts where their evolutionary stability becomes critical.

## Materials and methods

### *D. pseudotakahashii* field collections and laboratory lines

Female *D. pseudotakahashii* were obtained from 2 locations in New South Wales, 3 locations in south-eastern Queensland, and 6 (pooled) locations in northern Queensland [23] (S1 Table). All females were used to initiate isofemale lines. These collections yielded 7 female-biased (MK) lines from several south-east Queensland sites and a female-biased line from a New South Wales site that were initially unidentifiable due to the complete absence of male progeny but suspected to be *D. pseudotakahashii*. By introducing males from identified lines of *D. pseudotakahashii*, we determined that the female-biased lines were *D. pseudotakahashii*. Species identification was based on occasional male offspring identified through sex combs on tarsomeres I and II of the male foreleg and of the male terminalia (see [54]). Lines were screened for *Wolbachia* infection by PCR and RT-PCR (below). The incidence of female-biased lines across the populations was compared with a likelihood ratio analysis run in IBM SPSS Statistics 28 with probability determined using the Monte Carlo option.

To confirm that the female-biased lines were indeed *D. pseudotakahashii* and not a cryptic species, we used the *Drosophila* nuclear markers *Ddc* and *Pgi* [55] and the mitochondrial marker *CO1* [56] to screen a single individual from 3 confirmed *D. pseudotakahashii* lines that induce CI ($B116^{CI}$, $N51^{CI}$, and *Smith+*) and 5 female-biased lines ($N101^{MK}$, $B305^{MK}$, $B302^{MK}$, $B256^{MK}$, and $B289^{MK}$). DNA extractions using a 5% Chelex (Bio-Rad Laboratories, Gladesville, NSW, Australia; Cat. No. 142–1253; w/v in distilled water) and PCR conditions were performed as outlined in Richardson and colleagues [23]. PCR products were Sanger sequenced by Macrogen (Korea) and chromatograms were checked and edited manually in Finch TV v1.4.0 (Geospiza, Seattle, Washington, United States of America) and MEGA version 6 [57]. Sequences have been deposited in Genbank (accession number NZ_JAPJVH010000000).

Lines of *D. pseudotakahashii* used in crosses and experiments (S1 Table) were female-biased lines $N101^{MK}$ and $B302^{MK}$ and non-female biased lines $N51^{CI}$ and $B116^{CI}$. Females from all lines were infected with *Wolbachia* based on PCR characterization (see below). Because there were no naturally uninfected lines (see [23]), uninfected lines were generated for crosses from the isofemale line $TPH35^{CI}$ *(Town3+)* by treatment with 0.03% tetracycline (Sigma, Castle Hill, NSW, Australia) in cornmeal media for 1 to 2 generations (as outlined in Hoffmann and colleagues [58]). The derived *Wolbachia*-negative lines are designated with a "-" symbol as $TPH35^-$. Three of the female-biased lines ($N101^{MK}$, $B302^{MK}$, and $B305^{MK}$) and 1 non-female–biased line ($B116^{CI}$) were also treated with tetracycline as outlined above and their treated counterparts denoted by $N101^-$, $B302^-$, $B305^-$, and $B116^-$. The sex ratio of progeny was scored

after curing and the removal of *Wolbachia* infection was also verified via RT-PCR (see below). Lines were maintained in the laboratory on cornmeal media at 19˚C with a 12:12 L:D cycle and treated lines were allowed to recover in the absence of tetracycline for at least 2 generations before being used in experiments.

Lines (including those where the infection had been removed with tetracycline treatment) were monitored across time and their sex ratio was determined at F4 and F10 since setting up lines from the field. We collected at least 80 (range 80 to 390) per line to score sex ratio and compared results to an expectation of 50:50% male:female with chi-square tests.

## Crossing patterns and *Wolbachia* maternal transmission

We characterized the female-biased *D. pseudotakahashii* lines $N101^{MK}$ and $B302^{MK}$ by conducting a series of experiments investigating maternal transmission, CI, and sex ratio distortion. Experiments were conducted at 19˚C with a 12:12 L:D cycle.

We assessed maternal transmission of *Wolbachia* in *D. pseudotakahashii* female-biased lines in a two-part experiment using F4 and F5 individuals. In the first part, we crossed females from female-biased lines $N101^{MK}$ and $B302^{MK}$ with males from the $B116^{CI}$ and $N51^{CI}$ lines ($N = 15$). Of these, 4 lines were selected for part 2 of the experiment (including a line that produced both males and females denoted by *MKS* in which part 2 females were crossed with males from the CI lines $B116^{CI}$ and $N51^{CI}$ ($N = 84$) and males from the *MKS* line ($N = 10$) in addition to control crosses between the CI lines ($N = 12$)). For comparison, we also included crosses between the $N101^{MK}$ and uninfected $TPH35^-$ males ($N = 4$), and uninfected females and *MKS* males ($N = 4$), however replicate numbers were not high enough to investigate trends.

Crosses were set up when individuals were 4 to 7 days old, mating was observed after which males were removed and stored in ethanol. Females were provided with spoons containing cornmeal media and a brush of yeast paste to encourage egg laying. Spoons were scored for egg number and replaced every 24 h for up to 4 days. Twenty-four h after collection, eggs were scored for hatched and unhatched eggs. Given the comparatively low egg laying potential for this species [23], we used spoons where 7 or more eggs had been laid. Replicates that did not mate and had fewer than 7 eggs were removed from analysis. Progeny took approximately 18 days to develop; emerging adults were stored in 100% ethanol and sexed.

Hatch rates, development, and sex ratios among crosses were compared with nonparametric Kruskal–Wallis tests run in IBM SPSS Statistics 28.

## Long-term stability of the MK *Wolbachia* infection

Sex ratios of the $N101^{MK}$ and $B302^{MK}$ lines were scored approximately 68 generations after the lines were initiated. We also reassessed infection status and determined the infection densities for 10 males and 10 females from a subset of the lines (originally female-biased: $N101^{MK}$, $B305^{MK}$, $B302^{MK}$, $B289^{MK}$, $B256^{MK}$ and not female-biased: *Smith+*, $N51^{CI}$, $B116^{CI}$) using the Roche LightCycler 480 system and the strain-specific qPCR primers outlined below.

### *Wolbachia* strain detection and strain typing

A preliminary screen for *Wolbachia* infection was conducted for all field isofemale lines. DNA extractions were performed using the 5% Chelex-based method outlined above. Samples were screened for *Wolbachia* via RT-PCR using the *wsp_validation* primers [59,60]. This assay determines *Wolbachia* infection from the melting temperature ($T_m$) of the *wsp* validation PCR amplicons. High-resolution melt analysis on the Roche LightCycler 480 system produced a $T_m$

range from approximately 81.1 to 81.9°C for *D. pseudotakahashii*, including the female-biased lines.

To investigate the *Wolbachia* infection of the female-biased lines in more detail, we initially used the forward and reverse *coxA*, *hcpA*, *ftsZ*, *fpbA*, and *gatB* MLST primers [31] and *wsp_validation* primers [59,60] to screen a single individual from *N101^MK^*, *B302^MK^*, and *B289^MK^*. Conditions were as outlined in Richardson and colleagues [7], PCR products were sent to Macrogen (Korea) for purification, and Sanger sequencing chromatograms were examined and processed as outlined above using Finch TV v1.4.0 (Geospiza, Seattle, Washington, USA) and MEGA version 6 [57]. Sequences revealed the presence of double peaks for many of the MLST markers—notably absent in chromatograms of the *D. pseudotakahashii Wolbachia* strain *w*Pse generated by Richardson and colleagues [23]. Double peaks can be indicative of double infection [31] and although we initially used MLST as an established system that would allow links to be made to previous work, we conducted further sequencing analysis of the potential double infections using whole-genome sequencing, which is a preferable approach [35,61].

For general screening of field and experimental samples, we were unable to use established genotyping assays that distinguish CI and MK strains on the basis of melt analysis using *wsp_validation* primers (e.g., Richardson and colleagues [7]) because the Tm of the 2 strains in *D. pseudotakahashii* overlapped (81.1 to 81.9°C). Instead, we initially designed reverse primers to distinguish the 2 strains via standard PCR, based on alignments from the *gatB* MLST sequences we generated for the female-biased and non-female–biased lines (gatB_pt_MK_R: 5′-GTATCTATAATCGCTTGCATCCTC-3′ and gatB_pt_CI_R: 5′-GGCAACAAGTCAGG CTCA3′) and used these in conjunction with the existing *gatB* forward primers to produce bands of 373 bp and 388 bp that amplified the female-biased and non-female–biased strains, respectively. The PCR conditions for both primers were the same (3 min at 94°C; 37 cycles of 30 s at 94°C, 45 s at 71°C, 90 s at 72°C, and a final extension for 10 min at 72°C followed by holding at 4°C). A subset of the isofemale lines initiated from field collections including 98 lines from Brisbane, 11 from far northern Queensland, and 2 from Nowra were screened with these primers. To determine relative densities of the strains, we later used genomic sequences (outlined below) to design strain-specific primers specifically for use on the Roche LightCycler 480 system (MK strain: MK_F1; CCTTGTATTGAACTTCATCTTTGTTAC and MK_R1; GAACTTGTTTTACTTTATCACTTATCAC and CI strain: CI_Fc; TTCGATAAATAGAC TTTTAAACTCTGTA and CI_Rc; TTTTAGACAATCTTGATAATCTTGC). Primer-specificity was confirmed using several samples that were known to be singly infected with CI or MK and ensuring there was no cross amplification with non-strain–specific primers. The qPCR conditions for the CIc and MK3 primers were the same with an annealing temperature of 63°C. Using this system, we generated crossing point (Cp) values for MK and CI strain-specific markers (outlined below) in addition to the *Drosophila* universal primers targeting the ribosomal protein gene *L40* (RpL40) [23]. Densities were calculated based on methods outlined in [23] using the mean Cp value generated by 2 replicate runs and a standard deviation threshold of 2.5.

### *Wolbachia* whole-genome sequencing and assembly

We next sequenced whole genomes for the *N101^MK^* and *Smith+* genotypes using both short- and long-read technology. To produce short reads, total DNA was extracted from males ($N = 5$) and females ($N = 5$) of each genotype using the DNeasy Blood & Tissue kit following manufacturer protocol (Qiagen). Illumina libraries were prepared using the Nextera DNA Flex Library Preparation Kit (Illumina). Final fragment sizes and concentrations were

confirmed using a TapeStation 2200 system (Agilent), and the samples were indexed using IDT for Illumina Nextera DNA UD indexes. Pooled libraries were shipped to Novogene (Sacramento, California, USA) for sequencing on a partial Illumina NovaSeq lane, generating paired-end 150 bp reads. DNA was extracted from pooled individuals ($n$ = 10) using a DNeasy Blood & Tissue kit (Qiagen, Hilden, Germany). A continuous long-read library was prepared and sequenced using PacBio Sequel II technology by Berry Genomics (Berry Genomics Co., Beijing, China). PacBio sequencing yielded 8.91E+05 reads from $N101^{MK}$ and 8.22E+05 reads from $Smith+$, with a fasta size of 1.53E+10 and 1.57E+10 bp and approximate coverage of 50.96 and 133.65. The PacBio reads for the $N101^{MK}$ and $Smith+$ genotypes were assembled, and the *Wolbachia* contigs were extracted. Our assembly resulted in a final set of 314 scaffolds from $N101^{MK}$ with total length of 1.58E+06 bp and N50 of 19,011 bp, and 38 scaffolds from $Smith+$ with total length of 1.26E+06 bp and N50 of 62,575 bp. We polished the PacBio assemblies using the Illumina libraries and pilon v 1.23 [62] set to default parameters.

To evaluate the quality of the draft *Wolbachia* assemblies, we used BUSCO 3.1.0 [63] to search for homologs of the single-copy genes in the proteobacteria database. As a control, we performed the same search using the reference *w*Mel genome [64].

## Genomic analyses

To assess the genomic overlap and potential copy number variants (CNVs) in $N101^{MK}$ compared to $Smith+$, we aligned the Illumina reads for N101 and $Smith+$ to the draft $Smith+$ *Wolbachia* genome using bwa 0.7.17 [65] (the draft $Smith+$ *Wolbachia* genome was a single circular chromosome). Normalized read depth for each alignment was calculated over sliding 1,000 bp windows (1 to 1,000, 500 to 1,500, etc.) by dividing the average depth in the window by the average depth over the entire genome. The normalized read depth was plotted and visually inspected for regions with normalized depth different from 1. We capped the normalized depth of each window to 5 for readability.

We next searched for WO prophages in *w*Pse using the WO serine recombinase [29], which form 4 distinct WO prophage clades: sr1WO, sr2WO, sr3WO, and sr4WO [66]. Thus, we used BLAST to search for homologs to the serine recombinases observed in WOCauB3 (sr1WO), WOVitA1 (sr2WO), WOMelB (sr3WO), and WOFol2 (sr4WO) in our draft $N101^{MK}$ and $Smith+$ assemblies.

Recent research has identified CI-causing factors (*cifs*) associated with WO prophage in *Wolbachia* genomes [27,28,67], including individual SNPs that influence CI strength [68]. There are 5 characterized clades of *cif* loci (Types 1 to 5), which we denote using subscripts (e.g., $cifA_{wMel[T1]}$) (we use similar notation for *wmk* described below). We searched for the 5 *cif* types in *w*Pse genomes by BLASTing *cifs* from *w*Mel ($cifA_{wMel[T1]}$), *w*Ri ($cifA_{wRi[T2]}$), *w*No ($cifA_{wNo[T3]}$), *w*Pip ($cifA_{wPip[T4]}$), and *w*Stri ($cifA_{wStri[T5]}$) against our draft $N101^{MK}$ and $Smith$ + assemblies.

While the genetic basis of MK remains unknown, the *wmk* gene associated with the WO prophage region of non-MK *w*Mel kills male embryos when transgenically expressed in *D. melanogaster* [26]. We specifically searched for *wmk* by BLASTing *w*Mel *wmk* (WD0626) against our draft $N101^{MK}$ and $Smith+$ assemblies.

## Phylogenetic analyses

To extract genes for our phylogenetic analyses and identify homologs to known bacterial genes, we annotated our draft $Smith+$ genome and the public genomes of *w*Ana from *D. ananassae* [69]; *w*Au, *w*Ha, *w*Ri, and *w*No from *D. simulans* [34,70,71]; *w*Mel from *D. melanogaster* [64]; *w*NFa, *w*NFe, *w*NLeu, and *w*NPa from *Nomada* bees [72]; *w*Pip_Pel from *Culex*

*pipiens* [36]; and *w*Yak from *D. yakuba* [25] with Prokka v.1.11 [73]. To avoid pseudogenes and paralogs, we used only genes that were present in a single copy and with identical lengths in all analyzed sequences. Genes were identified as single copy if Prokka uniquely matched them to a bacterial reference gene. By excluding homologs that were not of equal length in our draft *Wolbachia* genomes, we removed all loci with indels across any of the included sequences. In total, 168 genes with a combined length of 136,545 bp met these criteria. We did not include *Wolbachia* infecting the $N101^{MK}$ genotype in our phylogenetic analyses since we could not confidently separate them.

With these 168 genes, we estimated a phylogram using RevBayes v. 1.1.1, following the procedures of Turelli and colleagues [35]. Briefly, we used a GTR + Γ model with 4 rate categories, partitioning by codon position. Each partition had an independent rate multiplier with prior Γ (1,1) [i.e., Exp(1)], as well as stationary frequencies and exchangeability rates drawn from flat, symmetrical Dirichlet distributions [i.e., Dirichlet(1,1,1...)]. The model used a uniform prior over all possible topologies. Branch lengths were drawn from a flat, symmetrical Dirichlet distribution, and thus summed to 1. Since the expected number of substitutions along a branch equals the branch length times the rate multiplier, the expected number of substitutions across the entire tree for a partition is equal to the partition's rate multiplier. Four independent runs were performed and all converged to the same topology. Nodes with posterior probability <0.95 were collapsed into polytomies. For additional details on the priors and their justifications, consult Turelli and colleagues [35].

We also estimated phylograms for $cifA_{[T1]}$, $cifA_{[T2]}$, and for *wmk*. We placed $cifA_{wPse\_N101}$ $_{[1\&2]}$ and $cifA_{wPse\_Smith}$ copies among Type 1, and $cifA_{wPse\_Smith+}$ and $cifA_{wPse\_N101}$ copies among Type 2 copies included in Martinez and colleagues [39]. We placed $wmk_{wPse\_Smith+[1\&2]}$ and $wmk_{wPse\_N101[1\&2]}$ among *wmk* copies included in Perlmutter and colleagues [26]. We used the same methods described above for the *Wolbachia* phylogram to produce these 3 additional phylograms.

## Segregation of MK suppression

**Generation of lines.** We performed crosses to determine the basis of MK suppression and generate lines expressing MK or a 50:50 M/F sex ratio for molecular analyses. Unmated females from the $N101^{MKS}$ and $B302^{MKS}$ lines were mated individually to $B116^-$ males, with 20 replicates per cross. The $N101^{MKS}$ and $B302^{MKS}$ lines (with "S" denoting suppression of MK) carried both the CI and MK *Wolbachia* strains but no longer showed any female bias, while the $B116^-$ line was cured of its CI *Wolbachia* strain and was not previously infected with the MK strain. Offspring from each replicate vial were scored for sex ratio, then unmated female offspring were pooled across replicates and mated to $B116^-$ males individually, with this process repeated for 2 backcrosses. Three replicates of the cross between $N101^{MKS}$ and $B116^-$ lines produced female-only offspring; we set up an additional 10 replicates of each line and continued to cross them to B116- males to test whether the female-only phenotype was maintained in the following generations. After backcross 2, we set up a third backcross, with separate sets of crosses with female-only lines and mixed sex lines, with 20 replicates each. Additionally, 60 replicate females each of the *N101* and *B302* lines from backcross 2 were isolated following mating within the line, then stored for molecular analysis after producing offspring (Fig 3A). In each generation, 20 replicates each of the $N101^{MKS}$, $B302^{MKS}$, and $B116^-$ lines (self-crossed) were set up as controls, with an expected 1:1 sex ratio each generation. In all crosses, flies allowed to mate for 3 days then transferred to new food to produce offspring for 4 days.

In a second experiment, we reverted female-only lines to produce both male and female offspring. Individual females from the female-only lines derived from $N101^{MK}$ and $B302^{MK}$ were

crossed to males from the $N101^{MKS}$ and $B302^{MKS}$ lines, respectively, with 10 replicates each. Female offspring were then individually mated to $N101^{MKS}$ and $B302^{MKS}$ males in the F1 and B1 generations, with 20 replicates per cross.

To test whether suppression of MK was associated with changes in *Wolbachia* density, we determined the infection densities of the CI and MK *Wolbachia* strains from backcrossed $N101^{MK}$ and $B302^{MK}$ lines that produced only females or both male and female offspring, as well as the original $N101^{MKS}$ and $B302^{MKS}$ lines. Fifteen females and 15 males (when present) from each group were screened using the Roche LightCycler 480 system and the strain-specific qPCR primers outlined below.

**Reassessment of crossing patterns in segregated lines.** We performed an additional set of crosses to test the ability of $B302^{MKS}$ males to induce CI with $B116^-$ and $B116^{CI}$ females. We also tested whether the $B302^{MK}$ line, which was reverted to MK as described above, induced late-acting MK. Crosses were performed as described above (see "Crossing patterns and *Wolbachia* maternal transmission"), with 30 replicates established per cross. We included crosses within each line for $B302^{MKS}$, B116$^{CI}$, and $B116^-$ as controls, which were expected to show high egg hatch and egg to adult viability with a 50:50 sex ratio.

**Molecular analysis of segregating lines.** To investigate potential genes driving suppression of the MK phenotype, we utilized the double-digest restriction-site–associated DNA sequencing (ddRADseq) protocol developed by Rašić and colleagues [74] to construct ddRAD libraries and genotype females for genome-wide SNPs. Females were those resulting from backcross 2 that were isolated and stored for ddRADseq following oviposition (see above). We separated females into 2 groups for each of the *N101* and *B302* lines; those that produced female-only offspring (*MK*) and those that produced both male and female offspring (<70% female, *MKS*). Lines producing >70% females or fewer than 15 total offspring were excluded due to phenotype ambiguity (since MK may be leaky). DNA from 24 females from each of the 4 groups was extracted using Roche High Pure PCR Template Preparation Kits (Roche Molecular Systems, Pleasanton, California, USA). A total of 96 *D. pseudotakahashii* females from across the 4 groups were grouped into 4 libraries and sequenced by NovogeneAIT Genomics (Singapore).

Sequence data were processed with Stacks v2.54 [75]. We used the *process_radtags* program to demultiplex sequence reads, discarding reads with average Phred score below 20. We used bowtie v2.0 [76] to align reads to the *D. takahashii* genome assembly GCA_018152695.1 [77] using very sensitive alignment. Genotypes were called with the *ref_map* program, with effective per-sample coverage of $24.5 \pm 7.1$X. We used the *Populations* program to calculate genome-wide heterozygosity and $F_{IS}$ for each group, omitting sites with any missing genotypes (-R 1) and retaining both monomorphic and polymorphic sites [78]. We ran *Populations* again to output a set of SNPs that were called in ≥95% individuals and when ≥3 copies of the minor allele were present [79]. We visually compared allele frequency plots of the remaining 26,810 SNPs for regions where genetic structure covaried with MK suppression across the *N101* and *B302* lines.

**Spread of MK suppression in mixed populations.** To investigate the spread of MK suppression, we set up mixed populations of $N101^{MK}$ or $B302^{MK}$ females (which produced female-only offspring) and $N101^{MKS}$ or $B302^{MKS}$ females (which produced both male and female offspring) and tracked sex ratios across generations. *MK* and *MKS* females were mated to $B116^-$ and $N101^{MKS}/B302^{MKS}$ males, respectively, then added to vials at ratios of 9:1 and 4:1 *MK*: *MKS*. We set up replicate vials of $N101^{MKS}$, $B302^{MKS}$, and $B116^-$ as controls which were expected to maintain a 50:50 sex ratio across generations. To test whether the *MK* lines continued to produce female-only offspring across generations, we also set up vials of $N101^{MK}$ and $B302^{MK}$ which were crossed to $B116^-$ males each generation. We set up 5 to 10 replicate vials

for each treatment and control and these were tracked and maintained separately. Each generation, lines that produced only females were maintained by adding 5 *B116⁻* males, otherwise males were not added. Lines were allowed to mate for 3 days and were then transferred to new food to produce offspring for 4 days. Lines were scored for sex ratio every generation and maintained for 5 generations.

## Supporting information

**S1 Fig. An estimated Bayesian phylogram for various Group-A (red) and Group-B (blue)** ***Wolbachia*** **strains.** *w*Pse *Smith+* is a Group-A strain and outgroup to a larger clade containing *w*Ha, *w*Ri-like, and *w*Mel-like strains. The 4 *Wolbachia* infecting *Nomada* bees (*w*NFe, *w*NPa, *w*NLeu, and *w*NFa) are outgroup to the clade containing *w*Pse *Smith+*. These Group-A *Wolbachia* diverged from Group-B *Wolbachia* (*w*Pip_Pel and *w*No) up to 46MYA (divergence time superimposed from Meany and colleagues [37]). The phylogram was estimated with 168 genes and a total of 136,545 bp. Nodes with posterior probability <0.95 were collapsed into polytomies. The sum of all branch lengths was fixed to one. Very small branch lengths (i.e., = /< 0.003) are excluded to improve figure readability. The data underlying this figure can be found in https://doi.org/10.26188/21892974.v1.
(TIFF)

**S2 Fig. Relative densities of the (A) CI and (B) MK** ***Wolbachia*** **strains in** ***Drosophila pseudotakahashii*** **lines after long-term laboratory culture.** Data points show densities in individual adults while vertical lines and error bars show medians and 95% confidence intervals. Individuals testing negative for a *Wolbachia* strain were excluded. The data underlying this figure can be found in https://doi.org/10.26188/21862119.v1.
(TIFF)

**S3 Fig. An estimated Bayesian phylogram for various** ***cifA*[T1] copies.** Identical samples were collapsed into a single tip, and nodes with posterior probability <0.95 were collapsed into polytomies. The root position is not known, but the tree was midpoint rooted for legibility. *cifA*[wPse_N101[2]] is identical to *cifA*[wPse_Smith+] and sister to the *cifA*[T1] set observed in the genome of the unnamed *Wolbachia* variant that infects gall wasp *Biorhiza pallida*. *cifA*[wPse_N101[1]], *cifA*[wMel], *cifA*[wRec], *cifA*[wSan], *cifA*[wTeis], and *cifA*[wYak] comprise a polytomy that is sister to a clade containing *cifA*[wHa] and *cifA*[wSh], assuming that the true root does not fall within this focal clade. We only report the posterior probability node support values that are less than 1. The data underlying this figure can be found in https://doi.org/10.26188/21892974.v1.
(TIF)

**S4 Fig. An estimated Bayesian phylogram for various** ***cifA*[T2] alleles.** Identical samples were collapsed into a single tip, and nodes with posterior probability <0.95 were collapsed into polytomies. The root position is not known, but the tree was midpoint rooted for legibility. *cifA*[wPse_Smith+] and *cifA*[wPse_N101] copies are identical and sister to *cifA*[wBic] from *Drosophila bicornuta*, assuming the true root does not fall in this clade. All nodes shown have a posterior probability support value of 1. The data underlying this figure can be found in https://doi.org/10.26188/21892974.v1.
(TIF)

**S5 Fig. An estimated Bayesian phylogram for** ***wmk*** **copies presented in Perlmutter and colleagues [26].** Identical samples were collapsed into a single tip, and nodes with posterior probability <0.95 were collapsed into polytomies. The root position is not known, but the tree was midpoint rooted for legibility. Node support values <1 are denoted. The data underlying this

figure can be found in https://doi.org/10.26188/21892974.v1.
(TIF)

**S6 Fig. Relative densities of the (A, C) CI and (B, D) MK *Wolbachia* strains in *Drosophila pseudotakahashii* lines following backcrossing.** Females from the (A, B) $N101^{MKS}$ line or (C, D) $B302^{MKS}$ line were crossed to $B116^-$ males for 3 generations. *Wolbachia* density was measured in the original lines and backcrossed lines that produced both male and female offspring (*MKS*) or female-only offspring (*MK*). Data points show densities in individual adults, while vertical lines and error bars show medians and 95% confidence intervals. Individuals testing negative for a *Wolbachia* strain were excluded. The data underlying this figure can be found in https://doi.org/10.26188/21862119.v1.
(TIF)

**S1 Table. Collection locations for *D. pseudotakahashii* populations used in this study.**
(DOCX)

**S2 Table. Infection types present in samples screened from lines originally expressing CI and MK phenotypes at F11.**
(DOCX)

**S3 Table. Private allele counts, observed heterozygosity ($H_O$), expected heterozygosity ($H_E$), and inbreeding coefficients ($F_{IS}$) for the female-only and mixed-sex phenotypes of the N101 and B302 lines.**
(DOCX)

**S4 Table. Genes in region suspected of containing suppressor gene(s).** The swept region is on contig NW_025323476.1, starting at position 3,321,074 and ending at 4,637,826. The region is 1,316,752 bp in length. All 212 SNPs within this region have MAF of less than 0.1, whereas outside the region, higher-frequency SNPs are common. There are 131 unique genes within the region, with 153 unique gene products.
(DOCX)

## Acknowledgments

The authors would like to thank Qiong Yang, Katie Robinson, Nancy Endersby-Harshman, and Tim Wheeler for molecular assistance. We also thank Jake Brown, Jackson Young, Courtney Brown, Torsten Kristensen, Andres Andersen, and Christian Danielsen for technical assistance. Dylan Shropshire and Julien Martinez provided comments that improved an earlier version of this work.

## Author Contributions

**Conceptualization:** Perran A. Ross, Brandon S. Cooper, Ary A. Hoffmann.

**Data curation:** William R. Conner, Thomas L. Schmidt.

**Formal analysis:** Perran A. Ross, William R. Conner, Thomas L. Schmidt, Ary A. Hoffmann.

**Funding acquisition:** Brandon S. Cooper, Ary A. Hoffmann.

**Investigation:** Kelly M. Richardson, Perran A. Ross, Brandon S. Cooper, Ary A. Hoffmann.

**Methodology:** Kelly M. Richardson, Perran A. Ross, Brandon S. Cooper, William R. Conner, Thomas L. Schmidt, Ary A. Hoffmann.

**Project administration:** Kelly M. Richardson, Brandon S. Cooper, Ary A. Hoffmann.

**Resources:** Brandon S. Cooper, Ary A. Hoffmann.

**Software:** William R. Conner, Thomas L. Schmidt.

**Supervision:** Brandon S. Cooper, Ary A. Hoffmann.

**Visualization:** Perran A. Ross, William R. Conner.

**Writing – original draft:** Kelly M. Richardson, Perran A. Ross, Brandon S. Cooper, Ary A. Hoffmann.

**Writing – review & editing:** Perran A. Ross, Brandon S. Cooper, Thomas L. Schmidt, Ary A. Hoffmann.

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
