## [Editor Report · Decision Letter 0]

7 Oct 2022

Dear Dr Hoffmann, 

Thank you for submitting your manuscript entitled "Hidden endosymbionts: A male-killer concealed by another endosymbiont and a nuclear suppressor" for consideration as a Research Article by PLOS Biology.

Your manuscript has now been evaluated by the PLOS Biology editorial staff, as well as by an academic editor with relevant expertise, and I'm writing to let you know that we would like to send your submission out for external peer review.

Once your full submission is complete, your paper will undergo a series of checks in preparation for peer review. After your manuscript has passed the checks it will be sent out for review. To provide the metadata for your submission, please Login to Editorial Manager (https://www.editorialmanager.com/pbiology) within two working days, i.e. by Oct 11 2022 11:59PM.

Kind regards,

Roli Roberts

Roland Roberts, PhD

Senior Editor

PLOS Biology

rroberts@plos.org

---

## [Decision Letter · Decision Letter 1]

23 Nov 2022

Dear Dr Hoffmann,

Thank you for your patience while your manuscript "Hidden endosymbionts: A male-killer concealed by another endosymbiont and a nuclear suppressor" went through peer-review at PLOS Biology. Your manuscript has now been evaluated by the PLOS Biology editors, an Academic Editor with relevant expertise, and by three independent reviewers.

In light of the reviews, which you will find at the end of this email, we are pleased to offer you the opportunity to address the [comments/remaining points] from the reviewers in a revision that we anticipate should not take you very long. We will then assess your revised manuscript and your response to the reviewers' comments with our Academic Editor aiming to avoid further rounds of peer-review, although might need to consult with the reviewers, depending on the nature of the revisions.

You'll see that the reviews are broadly positive about your study, but each reviewer raises a number of issues that will need to be addressed. Reviewer #1 is enthusiastic and their extensive comments only involve textual and presentational changes (as far as I could see). Reviewer #2 is disappointed that you could not tease out the full MK genome, makes some suggestions how this could be done, and wonders whether Wolbachia could be diploid (or other possibilities…). Reviewer #3 is also dissatisfied with the MK sequencing, and suggests long-read and bioinformatic solutions; s/he also requests a further analysis and has some textual and presentational issues.

IMPORTANT: I discussed the request from reviewers #2 and #3 for a full-length MK genome assembly with the Academic Editor. Overall, we think that while this would be "nice to have," it is likely that higher quality genome and comparative analyses (independent of assembly status) would satisfy these reviewers' concerns significantly, and we would not insist on a full assembly.

**IMPORTANT - SUBMITTING YOUR REVISION**

*Resubmission Checklist*

*Published Peer Review*

*PLOS Data Policy*

*Blot and Gel Data Policy*

Sincerely,

Roli Roberts

Roland Roberts, PhD

Senior Editor

PLOS Biology

rroberts@plos.org

REVIEWERS' COMMENTS:

Reviewer #1:

This paper is a holistic organismal investigation of the parameters that influence the existence of a double symbiont infection in flies in which the symbionts are CI-Wolbachia and MK-Wolbachia. I found the investigation largely comprehensive and the insights relevant to those studying MK in natural and laboratory populations. This paper stands on its own for documenting the co-existence of CI and MK Wolbachia combined with a suite of studies that contextualize this observation. While many individual pieces of data are not new, the combined approaches are a solid standard for the field to think about. Please find below my major and minor comments, all of which are important to update in the revision. 

Major comments.

1. This paper by Hurst et al. seems to be essential for the introduction and discussion of the paper, the latter of which could be more deeply expanded as it felt too short for such a wonderful set of results.

Which Way to Manipulate Host Reproduction? Wolbachia That Cause Cytoplasmic Incompatibility Are Easily Invaded by Sex Ratio-Distorting Mutants by Hurst et al. 

2. The abstract should mention the presence of wmk in the MK Wolbachia for contexual, genetic relevance to the field and potential genetic underpinnings of the trait. This makes the work more interesting to the rapidly developing area.

3. Add branch lengths to the phylogeny. Move the phylogeny earlier into the results for evolutionary reference and context relative to the rest of the study. 

4. Make a wmk and cifA phylogeny. Add other wmk homologs that do not kill males here too into the wmk tree. See below for related paper and wMel alleles. 

5. The quote from the paper (See below) is only partially true since mortality is elevated throughout development, but there is liekly embryonic lethality when comparing the CI control cross of 95% egg to larvae hatching versus several of the other crosses that show 80% egg to larvae hatching (MK female x CI male and CI x CI). Non-parametric ANOVA statistics for small sample sizes can be completed on all of these crosses and mortality stages to reveal where the effects. 

"This suggests that male killing is occurring, but at a later time point than expected based on studies in other species (e.g. D. pandora, [6])."

6. Rapid loss of MK section is very interesting, but it lacks a clear explanation in its current form. Im very curious about it. I see three crucial questions to be addressed in this section and the discussion. Add for assisting readers to think more broadly about this discovery. 

(a) What are the reasons for which this may occur so rapidly in the lab? What is failing in the MK strain to cause MK? Does lab acclimation or rearing media/conditions hide MK for some reason? I don't know. 

(b) Doesn't this phenomena also explain why MK could be relatively less common than other phenotype such as CI? Strains from the field that are maintained in the lab may hide the actual frequency of MK, if MK is not readily measured right away.

(c) Related to (b), Doesn't this phenomena also explain why some Wolbachia with MK genetic ability, like wMel, also not show MK in the lab? It struck me as an important, potential explanation for why wMel contains the wmk gene, but does not elicit MK.

7. wmk in N101: it is highly interesting, though not emphasized enough in this version of the paper, that the genetic evidence for the MK strain N101 yields near complete wmk identity to the male-killing wmk gene in wMel, whereas the CI strain wmk sequences are quite divergent. The authors can highlight this association in the results and enhanced discussion section on wmk. Also, they can evaluate if the CI wmk copies are more related to wmk-related sequences tested that did not yield MK in Perlmutter et al. 2019, mSystems, "See section on Divergent wmk homologs in wMel do not…" with divergent wmk homologous genes 508, 622, 623, and 255 that did not yield MK. Notably that study found only wmk from wMel is hared across all MK genomes to date, and it is the only one that kills males by transgenic expression in Dmel. 

8. The last sentence on page 12 strikes a tone that should be more balanced. e.g, "reflects anything about the genetic basis of MK in nature. Not only is wmk from wMel the only version of wmk shared across all MK genomes (some natural), but it is the distinguishing and causal gene of MK between related moth species that vary in MK or no MK. https://www.biorxiv.org/content/10.1101/2022.06.12.495854v2. It also seems probable or at least noteworth based on the sequence evidence presented here that the close wMel relative of wmk in the MK strain is the causal factor.

Minor comments:

1. Mention the other reasons for how/why MK symbionts spread (e.g., horizontal transfer of Wolbachia by dead siblings, resource reallocation, etc) here "such as through the avoidance of sib mating" in the intro.

2. Mention wmk presence and divergence here as a key result and for contextual relevance as readers will wonder about wmk in particular since it is the MK candidate supported by multiple genetic and host system studies (Flies and Moths): "We use molecular approaches to characterize the MK strain which differs for some genomic regions to the coinhabiting CI strain but is identical in other regions"

3. More information in T2 legend will help clarify if the loss/reduction of MK in F10 generation (From F4) is due to backcrossing with a non-MK strain or waning of the MK phenotype in laboratory-reared line. That's an important difference.

4. Under "Genomic analysis...." section, Smith+ is first mentioned here. What is this? Please add context. 

5. Are the cif types between Smith+ and N101 idential or not. Please mention in the results.

6. At "Our analyses also discovered..." Qualify what wmk is when it is first mentioned in results so reader knows the gene.

Nice study and kudos!

Reviewer #2:

All maternally-transmitted microbes that are successful in the long range in a population have to manipulate the reproductive biology of their hosts by favoring infected females for their own spreading and maintenance, or attenuate by taking over mutualistic functions ranging from higher host fitness, fecundity, pathogen protection or behavior. As a general but pivotal problem, however, rare endosymbiotic variants with the capacity to induce significant host phenotypes like sex ratios can be easily overlooked because they are overshadowed by dominating or even fixed variants in multi-infected hosts. Hence, we most likely only see currently only the tip of the iceberg and miss the full repertoire of microbes, strains and variants affecting that can affect host phenotypes.

In this manuscript, the authors have uncovered a male-killing (MK) Wolbachia strain variant in Drosophila pseudotakahashii flies that only works in combination with the fixed CI-inducing variant and not on its own. Interestingly, doubly-infected males mated to uninfected females induce strong CI, but when mated to mono-infected females they induce MK. This is a novel and exciting observation that will expand our conceptual understanding on multi-symbiont variant interactions and their population dynamics within an individual host. In addition, different to most MK Spiroplasma or Wolbachia doubly-infected males die later at their larval and not embryonic stage, which is quite unusual and will need further investigations on its mechanism - see below. 

By comparative genomics they found 17 diagnostic genes that are duplicated in the MK variant compared to the CI strain. It is a petty, however, the authors could not generate the complete MK genome of N101 with high confidence, which seems to be a technical issue since it is not feasible to separate both infections at the moment. Thus, the authors assume a double infection but unfortunately cannot provide full evidence. Did the authors try to isolate DNA from late and dying male larvae assuming the potential dominance of the MK variant over the CI variant at this stage? Hence, it would be worth checking ratios by diagnostic qPCR at different developmental stages in both sexes. Possibly, host-directed titer and/or tropism control of the MK variant via autophagy is less efficient in male larvae. Finally, can we rule out that Wolbachia is diploid? Alternatively, similar to the pathogenic wMel-Pop variant in D. melanogaster, this variant could over-replicate selectively in male larval tissues but not females explaining their late sex-specific mortality. The authors should at least consider and discuss these possibilities in the light of wMelPop and wMelCS and other potential explanations.

Finally, in response to the cost of rare males in natural populations the emergence of nuclear MK-suppressor was already earlier described in different insect associations, but their genetic basis and molecular mechanisms are still quite unclear or rudimentary. In this study, they isolated a by elegant backcrossing a dominant nuclear suppressor region of app. 1 Mb size that emerged in the lab spontaneously by having the capacity to rescued the MK phenotype, although the variant was still present. Future studies will hopefully uncover the actual responsive gene(s) in more detail.

Although the MK strain is transmitted maternally at high fidelity but not complete in the lab, it is found only at very low frequency in nature at 4%, that is a quite common prevalence for also other MK endosymbionts of insects. In future experiments - and not this current study since not the main focus here - it would be interesting to test if single infected females avoid doubly infected males in nature or in controlled mate choice assays, which would point to behavioral suppression/avoidance or multi-mating.

To sum up this study uncovers a new and exciting symbiotic model system to deepen our understanding on the complexity and genetic dynamics of both or even more partners on phenotype expression.

Reviewer #3:

This is an interesting account of i) presence of male-killing in a Drosophila as a coinfection ii) presence of suppression of male-killing that spreads through mixed culture from rare. The key messages that we derive are: a) PCR based screens for male-killers would fail in the presence of coinfection with similar symbionts that don't kill males. b) Male-killing can act at the larval stage in Drosophila c) Suppression is polymorphic in this system d) Suppression sweeps rapidly in laboratory mixed culture.

I think these are all well substantiated conclusions. I think c) and d) are probably the most exciting outcomes - in particular polymorphic suppression within a population is rather rare and presents opportunities for onward study. I'd argue a) was particular to Wolbachia male-killers with Wolbachia coinfection (heterospecific combinations would come out fine) - and to PCR screening for male-killers via sex biased presence, rather than phenotype based screening; thus whilst the point is well made and well taken, it is perhaps not the key driver to publish this in PloS Biology. b) (late male-killing) is interesting, but known for other male-killers (viral male-killing in tea tortrix moths, Spiroplasma in hemiiptera, and also Wolbachia introgression in Drosophila).

Important matters that may require some additional experimental work: 

1. Whilst I found the case for coinfection of two similar Wolbachia, one CI and one MK, to be well made, I felt the approach to genomic analysis of Wolbachia difference to be suboptimal. Yes, we see read depth changes that imply deleted regions, but the approach can't find gain of function compared to the single infection. Use of longer read technologies (either nanopore or PacBio) would enable this complete analysis. I felt the complete analysis was important as it would allow a) precise definition of the genome for the MK strain b) given the depth of background for this strain type, a good capacity to examine potential male-killing factors. It may also be possible to do this without extra sequencing: to extract novel regions belonging to the MK with bioinformatic approaches e.g blobtools identifying Wolbachia elements and then subtracting the known genome?

Important matters that require some extra analyses and modifications in text.

2. Having defined a genomic region for suppression, it seems straightforward to improve the reach of this paper to include a map of this region and predicted genes (even if onward analysis is left for a later paper).

3. The results of the PCR screen for the MK Wolbachia find a consistent association of presence with male-killing from field samples, which implies suppression is rare (or not expressed in flies taken from the natural environment). I felt this rarity deserved more contrast and discussion - in an ideal world this would be a population genomic screen of the population to ascertain the frequency of the variant associated with suppression, but that would be perhaps beyond this manuscript. Nevertheless, Further discussion of why rare would be worthwhile (usual theory explanation is suppression is costly in absence of MK so you get a polymorphic equilibrium).

Things I think would improve the manuscript but can be considered discretionary.

i) I found the abstract rather a hard read - needs to think about the non-specialist a bit. I could read it precisely after reading the manuscript but found a bit hard as a standalone. The first sentence is true but not perhaps the main point of the manuscript.

ii) Intro paragraph 2 - late male-killing also known in insects e.g. https://doi.org/10.1093/jhered/est052

iii) Page 4. Suppression allows CI to be revealed (as males now survive and express the CI) rather than evolve (although it does evolve phenotypically, it did not genetically).

iv) Table 1 could be tested for heterogeneity between samples (I suspect no evidence to reject homogeneity as frequency is low - but worth doing). Also, the text around this table doesn't quite mirror the data in the table in terms of total N and number of lines?

v) I'd probably move figure 2 to the supplementary as not essential to the more high profile points being made.

vi) Page 18 'genetic construct' sits awkwardly as has a meaning in functional genetics as an ectopic modification. Rephrase?

vii) P23 worth noting that suppression has been observed to spread readily in the field - so the lab here mirrors nature. See studies of both butterflies and lacewings.

viii) Polymorphic male-killer suppression is known in one other case, which should be referenced at some point. 10.1371/journal.ppat.1000987

Finally, this manuscript wasn't a very easy read in some place, partly because experiments are complex (fine) but also a bit because I think there are a lot of lab-specific names that the outside reader spends some mental energy remembering and working through. It was worth the effort to do so, but would suggest that the authors look at this to improve the clarity for the reader. e.g on page 11 we are introduced to Smith + but really have no introduction to this strain prior to this.

---

## [Editor Report · Decision Letter 2]

10 Jan 2023

Dear Ary,

Thank you for your patience while we considered your revised manuscript "Hidden endosymbionts: A male-killer concealed by another endosymbiont and a nuclear suppressor" for publication as a Research Article at PLOS Biology. This revised version of your manuscript has been evaluated by the PLOS Biology editors and the Academic Editor.

Based on our Academic Editor's assessment of your revision, we are likely to accept this manuscript for publication, provided you satisfactorily address the following data and other policy-related requests:

a) Please change the title to something more explicit/informative, and try to avoid punctuation. We suggest the following: "A male-killing Wolbachia endosymbiont is concealed by another endosymbiont and a nuclear suppressor"

b) Please address my Data Policy requests below; specifically, we need you to supply the numerical values underlying Figs 1AB, 2B, 3AB, S1, S2AB, S3, S4, S5, S6ABCD, either as a supplementary data file or as a permanent DOI’d deposition (e.g. as part of your Figshare deposition).

c) Please cite the location of the data clearly in all relevant main and supplementary Figure legends, e.g. “The data underlying this Figure can be found in S1 Data” or “The data underlying this Figure can be found in https://doi.org/XXXX”

We expect to receive your revised manuscript within two weeks. 

*Published Peer Review History*

*Press*

Sincerely,

Roli

Roland Roberts, PhD

Senior Editor,

rroberts@plos.org,

PLOS Biology

DATA POLICY:

Regardless of the method selected, please ensure that you provide the individual numerical values that underlie the summary data displayed in the following figure panels as they are essential for readers to assess your analysis and to reproduce it: Figs 1AB, 2B, 3AB, S1, S2AB, S3, S4, S5, S6ABCD. NOTE: the numerical data provided should include all replicates AND the way in which the plotted mean and errors were derived (it should not present only the mean/average values).

We require the original, uncropped and minimally adjusted images supporting all blot and gel results reported in an article's figures or Supporting Information files. We will require these files before a manuscript can be accepted so please prepare and upload them now. Please carefully read our guidelines for how to prepare and upload this data: https://journals.plos.org/plosbiology/s/figures#loc-blot-and-gel-reporting-requirements

DATA NOT SHOWN?

---

## [Editor Report · Decision Letter 3]

23 Jan 2023

Dear Ary,

Thank you for the submission of your revised Research Article "A male-killing Wolbachia endosymbiont is concealed by another endosymbiont and a nuclear suppressor" for publication in PLOS Biology. On behalf of my colleagues and the Academic Editor, Harmit Malik, I'm pleased to say that we can in principle accept your manuscript for publication, provided you address any remaining formatting and reporting issues. These will be detailed in an email you should receive within 2-3 business days from our colleagues in the journal operations team; no action is required from you until then. Please note that we will not be able to formally accept your manuscript and schedule it for publication until you have completed any requested changes.

Best wishes, 

Roli

Senior Editor

PLOS Biology

rroberts@plos.org